# Technology for Position Correction of Satellite Precipitation and Contributions to Error Reduction—A Case of the ‘720’ Rainstorm in Henan, China

**DOI:** 10.3390/s22155583

**Published:** 2022-07-26

**Authors:** Wenlong Tian, Xiaoqun Cao, Kecheng Peng

**Affiliations:** 1College of Computer, National University of Defense Technology, Changsha 410000, China; tianwl@nudt.edu.cn (W.T.); pengkc@nudt.edu.cn (K.P.); 2College of Meteorology and Oceanography, National University of Defense Technology, Changsha 410000, China

**Keywords:** extreme precipitation, satellite precipitation estimation, data fusion, image registration, position correction

## Abstract

In July 2021, an extreme precipitation event occurred in Henan, China, causing tremendous damage and deaths; so, it is very important to study the observation technology of extreme precipitation. Surface rain gauge precipitation observations have high accuracy but low resolution and coverage. Satellite remote sensing has high spatial resolution and wide coverage, but has large precipitation accuracy and distribution errors. Therefore, how to merge the above two kinds of precipitation observations effectively to obtain heavy precipitation products with more accurate geographic distributions has become an important but difficult scientific problem. In this paper, a new information fusion method for improving the position accuracy of satellite precipitation estimations is used based on the idea of registration and warping in image processing. The key point is constructing a loss function that includes a term for measuring two information field differences and a term for a warping field constraint. By minimizing the loss function, the purpose of position error correction of quantitative precipitation estimation from FY-4A and Integrated Multisatellite Retrievals of GPM are achieved, respectively, using observations from surface rain gauge stations. The errors of different satellite precipitation products relative to ground stations are compared and analyzed before and after position correction, using the ‘720’ extreme precipitation in Henan, China, as an example. The experimental results show that the final run has the best performance and FY-4A has the worse performance. After position corrections, the precipitation products of the three satellites are improved, among which FY-4A has the largest improvement, IMERG final run has the smallest improvement, and IMERG late run has the best performance and the smallest error. Their mean absolute errors are reduced by 23%, 14%, and 16%, respectively, and their correlation coefficients with rain gauge stations are improved by 63%, 9%, and 16%, respectively. The error decomposition model is used to examine the contributions of each error component to the total error. The results show that the new method improves the precipitation products of GPM primarily in terms of hit bias. However, it does not significantly reduce the hit bias of precipitation products of FY-4A while it reduces the total error by reducing the number of false alarms.

## 1. Introduction

Precipitation, as an essential part of the global water flux cycle, can connect the water and energy cycle on the earth [1], whose spatial and temporal distribution and motion trajectories play an important role in physical processes such as global atmospheric circulation and water balance [2,3,4,5]. Natural disasters such as floods, droughts, and mudslides can be caused by an uneven spatial and temporal distribution of precipitation. Secondary hazards caused by heavy precipitation events have become some of the most serious natural disasters in the world [6,7]. These natural disasters pose a serious risk to people’s lives and property.

The Intergovernmental Panel on Climate Change (IPCC) states in its Sixth Assessment Report that the impact of global warming caused by carbon emissions is irreversible and that the probability of extreme weather will increase as a result [8,9]. In July 2021, heavy rains hit various parts of the world. On 14 and 15 July, extreme precipitation affected Germany greatly and led to flooding that caused more than 50 deaths [10]. Influenced by the 6th Typhoon In-Fa, China’s Henan Province suffered extreme precipitation events [11]. The ‘720’ extreme precipitation event in Henan is well worth studying because this event records the strongest hourly precipitation since meteorological observations are available in mainland China, and it is also the largest cumulative precipitation in northern China since 1975 [12]. Thus, numerous scholars have conducted a lot of research work on the rainstorm event. Nie and Sun explored the mechanism of the development of this extreme precipitation event. They found that moisture was transported from the Western Pacific to Henan along three routes, which included the Western Pacific Subtropical High (WPSH), WPSH and the tropical cyclone In-Fa, as well as WPSH and the tropical cyclone Cempaka, which resulted in long-term and large-scale heavy rain in Henan [13]. Li et al. analyzed the generation and development of quasi-stationary mesoscale vortices in the early stages during heavy rainfall events [12]. Xia analyzed the characteristics and abnormality of atmospheric circulations of the heavy rainfall event and found that the South Asian high pressure enhanced the eastward extension and the subtropical high pressure was unusually strong and northward. They also discovered that, similar to Nie’s findings, the southeast airflow on the southwest side of the subtropical high pressure transported enough water vapor to Henan Province [14]. Zhang used the observations from CYSNSS to study the distribution and impact of floods in Henan [15]. Shi used observations from BeiDou/GNSS to analyze the relationship between extreme rainfall processes and the precipitation water vapor [16].

According to data from the National Meteorological Observatory, heavy rain continued for six days from 17 to 22 July and the heaviest precipitation occurred between 19 and 21 July (https://en.gmw.cn/2021-08/18/content_35091062.htm (accessed on 7 June 2022)). In Henan, there were 794 measuring stations with more than 100 mm of heavy rainfall from 08:00 on 19 July to 08:00 on 20 July. In Zhengzhou and Pingdingshan, there was 250 to 401 mm of very heavy rainfall, and five national meteorological observation stations in the province broke the extreme value of daily rainfall since meteorological records [17]. The total precipitation in Zhengzhou City in the 24 h from 8:00 on 20 July to 8:00 on the 21st was 624.1 mm, which is equivalent to half of its annual precipitation. The peak hourly precipitation reached 201.9 mm/h at 9:00 on 20 July, which broke the strongest hourly precipitation since meteorological observations were available in mainland China [12,18]. In addition, since Henan Province is one of China’s major grain-producing and populous provinces, devastating disasters such as floods and urban inundation caused by extreme precipitation have a profound impact on agricultural production, human livelihoods, and economic development [19]. At the end of 2 August, the devastating flooding it caused had killed 302 people and affected more than 14.53 million people in 150 county-level areas. According to official data, more than 1.09 million hectares of crops have been damaged in Henan Province, more than 30,600 houses have collapsed, and the cumulative economic loss has exceeded CNY 120 billion [20,21]. Therefore, the observation and prediction of extreme precipitation weather are crucial.

The observation types of precipitation mainly include rain gauge, ground-based radar, satellite estimation, and so on [22]. Rain gauge and ground-based radar are traditional observation methods [23]. The rain gauge is the most direct and accurate observation method [24,25]. However, its spatial resolution is low due to its limited observation position, inflexible observational means, and more significant influence by station distribution [26]. Ground-based radar is gradually applied to precipitation observation with ground remote sensing technology development, which has solved the above problems to a certain extent. It is now an important tool for precipitation observation due to its high spatial and temporal resolution [27]. It has an excellent performance in observing slight and medium-scale precipitation, but the high cost and the shortcomings of terrain make it difficult to follow large-scale precipitation, which creates uncertainty in radar precipitation estimates [28,29]. Meteorological satellites specifically for remote precipitation measurement are created to solve the problem of traditional observation methods’ spatial limitations with the development of remote sensing technology and the emergence of high-resolution satellites [30]. They use active or passive remote sensing to invert precipitation, provide high-resolution precipitation observation data for the world and regions, make up for the uneven distribution of surface rain gauge stations and complex maintenance, and make up for the deficiencies of ground-based radar that are limited by terrain [31].

At this stage, the mainstream precipitation satellite is Global Precipitation Measurement (GPM). GPM is the successor to the Tropical Rainfall Measuring Mission (TRMM), which inherits the TRMM’s ability to detect large-scale and mesoscale precipitation. It can accurately detect light precipitation because it is equipped with a more advanced spaceborne dual-frequency Precipitation Radar (DPR) and GPM Microwave Imager (GMI) [1,31]. Since its release, the accuracy of Integrated Multisatellite Retrievals for GPM (IMERG) has been studied by numerous scholars [1]. For example, Gaona used rain gauge data to assess IMERG products released in the Netherlands for one year [32]. IMERG has been positively evaluated in terms of precision and accuracy in detecting precipitation compared with the precipitation products of TRMM. FY-4A is a new generation of China’s geostationary meteorological satellite launched in 2016. Since its release, it has provided China and the world with different data products, including quantitative precipitation estimation (QPE) (nsmc.org.cn (accessed on 7 June 2022)). Ren evaluated the summer QPE of FY-4A using rain gauges and found that it presented poor accuracy over the complex terrain of western China, and established cloud classification thresholds for FY-4A with dynamic clustering methods to identify convective clouds, which eventually improved the QPE [33]. In fact, few studies have been conducted on the QPE product of FY-4A, especially its performance in extreme weather and comparison with mainstream satellite precipitation products.

Due to the high spatial and temporal resolution and comprehensive coverage, satellite precipitation products have become an important data source for climate research and flood warnings [34,35]. However, the use of satellite precipitation estimation alone does not yet yield satisfactory results in hydrological and weather forecasting, even though it has good application potential [36]. Studying their data errors is essential to use satellite precipitation products better. At this stage, a great deal of work has been done to improve the numerical intensity of the precipitation. Multimodal data fusion has recently become a popular method for obtaining more accurate precipitation products [37,38]. In general, rain gauge data are often used as an additional data source to correct satellite precipitation estimates [39,40]. Wu improves quantitative precipitation estimation by fusing satellite precipitation with rain gauge station data using a combination of convolutional neural network (CNN) and long short-term memory network (LSTM) methods [41]. Zhou proposed an integrated methodology from precipitation to flow simulation in a data-scarce basin and used a small number of stations to bias-correct six satellite precipitation products to improve the data accuracy of the satellite precipitation products [42]. Le presented an efficient approach based on a combination of the convolutional neural network and the autoencoder architecture to correct the pixel-by-pixel bias for satellite products using APHRODITE, a gridded precipitation product generated from rain gauge observations from thousands of stations in Asia [43]. However, in addition to the numerical intensity of the product, the accuracy of the product’s geographical distribution position information should also be paid attention. Due to satellite inversion estimation algorithms and remote sensing observation errors, as well as the inherent high spatial and temporal variability of continuous moving weather systems such as precipitation, satellite estimates of the position and shape of precipitation events are prone to errors [44]. In some applications, such as data assimilation in hydrological models or numerical weather models, Lopez claimed that detecting the correct position of precipitation events is just as crucial as detecting their intensity [45]. The position error of the numerical weather prediction model has been considered in the forecast validation. In order to reduce the position errors, Hoffman et al. (1996) proposed a feature correction and alignment technique (FCA), which used a variational approach to solve a nonlinear least squares estimation problem with side constraints to vary the displacement and amplification errors of the prior background field until usable observations are available [46]. Based on the FCA method, Grasssortti fused radar and satellite precipitation estimates and performed the ideal experiment for Typhoon Andrew, using the precipitation observed by radar to adjust the precipitation estimates from SSM/I. The adjusted SSM/I precipitation estimates can better fit radar observations and satisfy other constraints [40]. Besides, FCA is widely used in variational assimilation. THOMAS applied displacement vectors to the 3D WRF field and improved the prediction of typhoon position using FCA [47]. To improve data assimilation, Bezley and Mandel (2008) combined image warping with the Ensemble Kalman Filter (EnKF), established a wildfire model, and registered two images through the image registration method [48]. This technique can be applied to position corrections to satellite precipitation data. Camille improved the loss function optimization method and applied it to rainfall events during the monsoon season in southern Ghana, Africa, with good results [44]. In general, there are few studies on position correction for precipitation. Many studies about position correction are directed at improving the data assimilation techniques, while even fewer studies are conducted on simply improving the satellite precipitation estimates. At present, most of the studies to improve satellite precipitation estimates are directed to the numerical intensity of precipitation; so, it is important to study the research on the position correction technique for satellite precipitation estimates.

Camille only confirmed the usefulness of the position correction method for light precipitation events due to the sparse observations from rain gauge stations and generally low precipitation in Africa, and its performance in heavy rainfall events was not proved [49]. In fact, the application of the position correction method is much more critical in heavy rainfall events. Furthermore, the contribution of the error components to the total error was not analyzed in Camille’s study, and the mechanism for improving the satellite error by the position correction method is unclear. In addition, the current mainstream satellite precipitation products are from TRMM and GPM [1,31], while there are few studies on the secondary precipitation products (quantitative precipitation estimation, QPE) of the FY-4A, a new generation of Chinese geostationary meteorological satellite launched in 2016 [33]. As a result, evaluating QPE and comparing it to precipitation products from GPM are both urgent tasks that must be done. In order to solve the above problems, in this paper, the observations from surface rain gauge stations are used to correct the position of satellite precipitation estimates, based on the idea of registration and warping in image processing [50]. Using the extreme precipitation event ‘720’ in Henan Province as an example, the performance of the satellite precipitation products FY-4A and IMERG before and after correction is evaluated. Further, the improvement of the new method on the errors of different satellite precipitation products in the extreme precipitation event is investigated. An error decomposition model is used to evaluate and analyze satellite precipitation data to investigate the contribution of each error component to the total error reduction after position correction.

The structure of the paper is as follows. Section 2 describes the rain gauge stations and satellite precipitation products, image registration and warping methods, and error decomposition models, respectively. Section 3 introduces the characteristics of the satellite precipitation estimates before and after the correction and analysis of the contribution of each component error. Section 4 discusses the performance comparison of different satellite products as a whole and the improved contribution of the error component to the total error after position correction; it also discusses the shortcomings and limitations of the method. The conclusions are given in Section 5.

## 2. Data and Methods

### 2.1. Data

#### 2.1.1. Surface Rain Gauge Station

The observation data from surface meteorological stations serve as the primary data source for some related research projects, including work on climate models, monsoon variability, and extreme weather disasters. Hourly data from national surface stations in China include hourly observations of the temperature, pressure, humidity, wind speed, and precipitation. The measured data recorded by the surface rain gauge stations can genuinely reflect the basic precipitation situation in the vicinity of the rain gauge stations. Before the year 2000, rain gauge station data in China were broken down into two categories: artificial rain gauge regular observation and self-recording tipping bucket or siphon rain gauge observation. In the 21st century, automatic rain gauge stations gradually replaced the original telemetry self-recording precipitation stations in surface meteorological stations across the country, making it possible to obtain hourly resolution precipitation data. The current hourly rain gauge station data are derived from the observations of automatic rainfall sensors [51]. The surface rain gauge data used in this paper come from the hourly observation data of the National Meteorological Science Data Center of the China Meteorological Administration, which publishes the hourly observation data for nearly 7 days in real time with a time resolution of 1 h. After quality control, the actual rate of each element of data exceeds 99.9% and the correct rate of the data is close to 100% (https://data.cma.cn/) (accessed on 7 June 2022). The surface rain gauge data used are from 19 July 2021 to 21 July 2021, which is the main impact period of the ‘720’ extreme precipitation event in Henan. The longitude range of the study area we chose is 110.95–115.95° E and the latitude range is 31.45–36.45° N, to conform to the satellite grid. The rain gauge data are collected from 179 national surface stations in the region.

#### 2.1.2. IMERG Precipitation Products

IMERG is a new generation of Integrated Multisatellite Retrievals of precipitation data of the Global Precipitation Measurement Program (GPM). It uses the satellite precipitation retrieval algorithm to organically fuse the data provided by all satellite sensors on the GPM. IMERG’s tertiary products have three precipitation products: Early Run, Late Run, and Final Run. The Early Run product is obtained after the IMERG generation system runs once in the real-time stage and is generally released 6 h after the observation data. The Late Run product can be received after rerunning, generally released 16 h after the observation. The difference is that only the forward propagation algorithm in the cloud vector propagation algorithm is adopted in the Early Run, while the backward propagation algorithm is also used in Late Run base on the Early Run. The Final Run product is generated by correcting the Late Run error using more sensors and monthly average station data from the Global Precipitation Climate Center (GPCC), which has a 3-month latency [52,53].

This paper intends to use the Late Run product for research and use the Final Run product as a reference for comparison. Both precipitation products are derived from the GPM’s third-level V6 product, IMERG, with a temporal resolution of 0.5 h and a spatial resolution of 0.1° × 0.1°, in mm/h. Data can be downloaded from https://disc.gsfc.nasa.gov/datasets (accessed on 7 June 2022).

#### 2.1.3. QPE Products of FY-4A

FY-4A is a new generation of geostationary meteorological satellite launched by China in 2016, carrying a variety of meteorological remote sensing observation instruments, including Advanced Geostationary Radiation Imager (AGRI), Geostationary Interferometric Infrared Sounder (GIIRS), Lightning Mapping Imager (LMI), and Space Environment Monitoring Instrument Package (SEP). The data used in this paper are from AGRI’s secondary-level product, quantitative precipitation estimation (QPE). The time resolution of QPE is available in 15 min, 1 h, 3 h, 6 h, and 24 h options. To match the time resolution of surface rain gauge stations, QPE with a time resolution of 1 h is selected in this paper. The original spatial resolution of QPE is 4 km in mm [54].

In this paper, the IMERG and QPE precipitation products are collectively referred to as satellite precipitation estimates (SPEs).

### 2.2. Methods

#### 2.2.1. Image Registration and Warping

Image registration is a matching technique in image processing that transforms two given images into each other by finding a spatial map [50]. The method has flourished in the field of medical imaging for studying changes in patient lesions [55,56,57]. It has recently been used in the matching of remote sensing images [58,59]. The original method, as implemented, finds a spatial mapping by taking the pixel values of two images to be registered as input. In meteorology, two fields can also be regarded as images to be matched, such as the precipitation fields. Define the two fields as fixed image *p* and moving image *q* on the domain $D\subset R^{2}$. A mapping domain ϕ can be found through image registration, such that, ∀(x,y)∈D,
(1)p(x,y)≈q∘ϕ(x,y)
$\phi :\left(x,y\right)\in R^{2}\mapsto \left(\phi_{x}\left(x,y\right),\phi_{y}\left(x,y\right)\right)\in R^{2}$ is often represented by a warping vector field *M* (from now on, referred to as the mapping function). *M* represents the set of warping vectors from the pixels of *p* to the pixels of *q*. ϕ=I+M, where *I* is the identity function, I:z↦z [60]. However, there can be several *M* satisfying the above equation, especially in regions without precipitation. Therefore, three weak constraints need to be defined for the optimal mapping [44]: (2)$$\begin{array}{c} M\approx 0\\ \nabla M=\left(\begin{array}{cc} \frac{\partial M_{x}}{\partial x} & \frac{\partial M_{x}}{\partial y}\\ \frac{\partial M_{y}}{\partial x} & \frac{\partial M_{y}}{\partial y} \end{array}\right)\approx 0\\ \nabla \cdot M=\frac{\partial M_{x}}{\partial x}+\frac{\partial M_{y}}{\partial y}\approx 0 \end{array}$$
The above constraints indicate the optimal *M* as small, smooth, and as divergent-free as possible.

Users need to select points in the two images that should map to each other manually in classical registration. We use the automatic registration method described by Ref. [48] that does not require any manual point-by-point input. The objective problem can be expressed as an optimization problem [61]: (3)$$\underset{M}{\arg\min}L\left(p,q,M\right)=\underset{M}{\arg\min}\left(L_{o}\left(p,q\circ \left(I+M\right)\right)+\lambda L_{b}\left(M\right)\right)$$
where q∘(I+M)) represents the mapping function acting on *q* and the loss function $L_{o}$ represents the similarity of the two images: (4)$$L_{o}\left(p,q\circ \left(I+M\right)\right)=∥p-q\circ \left(I+M\right)∥$$
The loss function $L_{b}$ represents weak constraints on the mapping and λ is the regularization parameter: (5)$$\lambda L_{b}\left(M\right)=\lambda_{1}∥M∥+\lambda_{2}∥\nabla M∥+\lambda_{3}∥\nabla \cdot M∥$$
where $\lambda_{1}$, $\lambda_{2}$, and $\lambda_{3}$ are regularization parameters for different weak constraints. ∥·∥ is the $L^{2}$-norm. The solution procedure of the optimization problem (Equation 3) using the idea of Ref. [44] is described with more details in Appendix A. In fact, there are other approaches to solving the optimization problem as well as the one used in this paper. For example, the optimization problem was solved by optimizing each node of *M* in Ref. [48].

When the optimal *M* is found through the registration method, the warped image *q* can be obtained immediately by Formula (Equation 1). The registration and warping method using rain gauge station data and satellite precipitation estimates is shown schematically in Figure 1.

#### 2.2.2. Satellite Data Evaluation Indicators

In order to evaluate and compare satellite precipitation estimates and quantify the effect of position correction on their improvement, several indicators are selected in this paper, which are commonly used in satellite precipitation research [53,54,62]. We evaluate the precision of satellite precipitation detection, the accuracy of satellite precipitation estimates, and the correlation between satellite and rain gauge data. The correlation coefficient (CC), whose ideal value is 1, reflects the degree of linear correlation between satellite precipitation and gauge observations and is a crucial indicator of how well the two are fitted. Root mean squared error (RMSE) and mean absolute error (MAE) are used to express the degree of dispersion between SPEs and rain gauge station data, whose optimal values are both 0. In comparison with MAE, RMSE gives larger errors more weight, making it more sensitive to SPE error dispersion [36]. MAE is applied to indicate the mechanism of improving the satellite precipitation error component by the position correction method. The relative deviation (RB), which measures the degree of systematic deviation of the SPE, can be used to analyze the error trend between the SPE and the measured data from the rain gauge station; its ideal value is 0. The probability of detection (POD) is used to represent the ability of satellite precipitation data to capture actual precipitation events, whose optimal value is 1. POD represents the ratio of the estimated hits of the observation satellite to the hits of the rain gauge. Regarding rain gauge data as the benchmark, the POD can reflect the detection performance of the satellite precipitation estimation. The false alarm rate (FAR) is the ratio of the number of false alarms in satellite precipitation estimates to the total number of detections, whose optimal value is 0; it can reflect the false alarms of precipitation events in satellite precipitation data. The critical success index (CSI) comprehensively considers the hits and false alarms of satellite precipitation data and reflects the true ability of satellite precipitation data to monitor actual precipitation events, whose optimal value is 1. The above evaluation index formulas are shown in Table 1.

#### 2.2.3. Error Decomposition Method

According to the method of Tian [63], an error decomposition model can be established to analyze the error of satellite precipitation estimation and the contribution of warping methods to error improvement. The total error (TE) of satellite precipitation estimation is decomposed into three components by the model, which are the biases of hit precipitation (HB), missed precipitation (MP), and false precipitation (FP); these come from hit precipitation events, missed precipitation events, and false precipitation events, respectively. A hit precipitation event indicates that the satellite and the ground station detected a precipitation event simultaneously. A missed precipitation event indicates that the satellite did not detect a precipitation event, but the surface station has a good precipitation record. A false precipitation event indicates that precipitation is detected by the satellite but not by the surface rain gauge stations [64].

Given a precipitation field $R(\vec{x},t)$, one can derive a binary-valued precipitation event mask $P(\vec{x},t)$: (6)$$\begin{array}{c} P\left(\vec{x},t\right)=\left\{\begin{array}{cc} 1 & \;\mathrm{if}\;R(\vec{x},t)>0\\ 0 & \;\mathrm{if}\;R(\vec{x},t)=0\;\mathrm{or}\;\mathrm{null}\; \end{array}\right. \end{array}$$
We consider less than 0.1 mm/h to be no precipitation because light precipitation is difficult to detect.

We set $P_{1}$ as the precipitation event mask with event 1 and $P_{\bar{1}}$ as the Boolean complement of the binary masks $P_{1}$, which means $P_{\bar{1}}$ is 0 when $P_{1}$ is 1. Event 1 is the surface rain gauge station data and event 2 is the satellite precipitation estimation in this section. We can define hit mask $P_{12}$, miss mask $P_{1\bar{2}}$, and false mask $P_{\bar{1}2}$, respectively, as
(7)$$\begin{array}{c} \left\{\begin{array}{c} P_{12}=P_{1}\times P_{2}\\ P_{1\bar{2}}=P_{1}\times P_{\bar{2}}\\ P_{\bar{1}2}=P_{\bar{1}}\times P_{2} \end{array}\right. \end{array}$$
It is easy to find that the above 3 masks are independent and orthogonal to each other: (8)$$\begin{array}{c} P_{12}\times P_{1\bar{2}}=P_{12}\times P_{\bar{1}2}=P_{1\bar{2}}\times P_{\bar{1}2}=0 \end{array}$$
Then, TE, HB, MP, and FP can be defined: (9)$$\begin{array}{c} \left\{\begin{array}{c} TE=R_{2}-R_{1}\\ HB=\left(R_{2}-R_{1}\right)\times P_{12}=HB_{2}-HB_{1}\\ MP=-R_{1}\times P_{1\bar{2}}\\ FP=R_{2}\times P_{\bar{1}2} \end{array}\right. \end{array}$$
where $HB_{2}$ and $HB_{1}$ denote the precipitation events hit in satellite precipitation estimation and rain gauge station, respectively. Since the three error components are relatively independent,
(10)$$\begin{array}{c} TE=HB+MP+FP \end{array}$$

## 3. Results and Analysis

### 3.1. Precipitation Field Image Registration and Warping

The position correction method based on the idea of image registration and warping is studied in this paper using the ‘720’ extreme precipitation in Henan as an example. The maximum precipitation time point for this extreme weather process, according to statistics, is at 14:00 on 20 July. Figure 2 shows that, in comparison to the stations, the precipitation fields of the three SPEs are distributed eastward; especially, the main precipitation area of FY-4A is 1° eastward. It can be seen that the SPEs have apparent positional deviations relative to the surface rain gauge data. In addition, there is also a significant error in SPEs in terms of the numerical intensity of the precipitation. In the vicinity of (113.5° E, 35° N), there are many precipitation stations with precipitation within 1 mm/h, but the SPEs do not detect it because it is difficult for satellites to observe light precipitation and the inversion algorithm is difficult to deal with light precipitation during extreme precipitation. The maximum of the precipitation observed from stations is 103.4 mm/h at the Kaifeng station, but there is little distribution over 50 mm/h in SPEs. The overall distribution of the IMERG final run product best fits the observed distribution of precipitation at the station and its maximum precipitation is the highest of the three SPEs, with several precipitation areas exceeding 50 mm/h. The IMERG late run is not much different from the final run but the numerical intensity is smaller, with the maximum value below 50 mm/h. The final run is gauge-adjusted at a monthly scale with the GPCC gauge product, whose performance is the best in terms of accuracy. The fit of FY-4A performs the worst among the three, with low and high precipitation values that are not accurately inverted. In general, satellite precipitation estimates have significant numerical inaccuracies and positional biases, resulting in significant non-Gaussianity in the data. Satellite data errors can affect climate studies and make numerical weather forecasts unreliable. We perform registration and warping processing on all three satellite products to obtain more accurate SPEs. For the registration at different moments of this example, Equation (Equation 3) can be extended as
(11)$$\underset{M_{i}}{\arg\min}L\left(\left[\begin{array}{c} p_{1},q_{1},M_{1}\\ p_{2},q_{2},M_{2}\\ \ldots \\ p_{t},q_{t},M_{t} \end{array}\right]\right)=\underset{M_{i}}{\arg\min}\left(∥\left[\begin{array}{c} {\tilde{p}}_{1}-{\tilde{q}}_{1}\circ \left(I+M_{1}\right)\\ {\tilde{p}}_{2}-{\tilde{q}}_{2}\circ \left(I+M_{2}\right)\\ \ldots \\ {\tilde{v}}_{t}-{\tilde{u}}_{t}\circ \left(I+M_{t}\right) \end{array}\right]∥+\lambda_{1}∥\left[\begin{array}{c} M_{1}\\ M_{2}\\ \ldots \\ M_{t} \end{array}\right]∥+\lambda_{2}∥\left[\begin{array}{c} \nabla M_{1}\\ \nabla M_{2}\\ \ldots \\ \nabla M_{t} \end{array}\right]∥+\lambda_{3}∥\left[\begin{array}{c} \nabla \cdot M_{1}\\ \nabla \cdot M_{2}\\ \cdots \\ \nabla \cdot M_{t} \end{array}\right]∥\right)$$
where $\lambda_{1}$, $\lambda_{2}$, and $\lambda_{3}$ in this paper are 0.1, 1, and 1, respectively [65].

We treat interpolated surface rain gauges (Figure 2a) as the fixed image *p* and three SPEs (Figure 2b–d) as the moving image *q*, respectively. Taking *p* and *q* as input to the automatic registration algorithm, the image warping vector field is obtained through iterative processing. The warping vector field is defined on the warping mesh $D_{i}$, which is applied to the image *q* to obtain the moved image. Data preprocessing is required before image registration. Firstly, light precipitation less than 0.1 mm/h needs to be removed because it is difficult to observe and can lead to significant uncertainties. In the second step, zero precipitation regions need to be added around the input images *q* and *p*. The original 50 × 50 grid is expanded into a 65 × 65 grid, with zero precipitation in the expanded portion. The problem of nearby minimization constraints is avoided by adding a zone of zero precipitation at the boundary. The longitude range of $D_{i}$ is currently 110.25–116.75° E, with a latitude range of 30.75–37.25° N.

After preprocessing, Figure 3 is obtained by satellite precipitation estimation after image registration and warping processing. The precipitation area of the IMERG final run after warping moves westward, as can be seen in the first row of Figure 3, and the precipitation center coincides with the rain gauge stations. The fit between SPEs and rain gauge stations has improved dramatically. The main direction of the warping vector is from southeast to west, as shown in the diagram. The situation for the IMERG late run is similar to that of the final run, with the precipitation area moving westward. FY-4A’s condition has improved slightly, with the precipitation area covering more stations and the precipitation center being closer to the center of precipitation observations from the stations. In general, image registration and warping techniques correct the position of SPEs to bring them closer to rain gauge stations.

### 3.2. Statistical Analysis

The ‘720’ extreme rainstorm event in Henan is mainly concentrated from 19 July to 21 July 2021. We performed hour-by-hour image registration and warping of SPEs for this period. To reduce the error in the analysis, we evaluated the SPEs’ error using rain gauge station data as the actual value. We used nearest-neighbor interpolation to interpolate satellite precipitation estimates before and after position correction to rain gauge stations to avoid errors caused by light precipitation in areas without precipitation caused by linear interpolation.

The maximum precipitation period of the ‘720’ Henan extreme precipitation event is from 14:00 to 21:00 on 20 July, as shown in Figure 4, with the maximum value at 14:00. Furthermore, the IMERG SPEs’ time series curve fits the station curve better. FY-4A was poorly fitted to the station curve with significant errors, especially during the period of 8:00–14:00 on the 21st. In addition, it can be found that the location correction does not significantly improve the numerical intensity of the precipitation.

Table 2 shows that after position correction by the warping technique, the RMSE, MAE, RMSE_gird, and RMSE_gird of the three SPEs are significantly reduced, demonstrating that position correction can reduce the SPEs’ error. It is worth noting that, while position correction can help reduce the FY-4A’s error to some extent, it still has a high error. FY-4A cannot accurately invert the light precipitation value or the high precipitation value in extreme precipitation. Surprisingly, the final run’s RMSE is lower than the late run before position correction but higher than the late run after position correction. This means that image warping has better correction performance for the late run. We believe this is because the final run product used more sensors and GPCC’s monthly average station data for error correction, resulting in the rain gauge stations having a limited effect on it. The correlation coefficient (CC), average position error (APE), and regression coefficient (RC) for SPEs can all be interpreted in the same way.

To study the distance relationship between SPEs and rain gauge stations, we define the distance between the positions of the precipitation center (maximum precipitation) as the position error. Firstly, SPEs are linearly interpolated to each rain gauge station. Then, select the maximum value at each time. The distance error index in this paper is the distance between this point and the maximum value of the rain gauge stations at the same time.

Table 2 and Figure 5 show that FY-4A has the most significant position error, while the final run has the smallest. The maximum errors of the three are concentrated between 5:00 and 12:00 on the 21st. Their position errors are significantly reduced after position correction. Table 2 shows that the late run satellite precipitation product has the smallest average position error after position correction, with an improvement of 46%.

Table 2 and Figure 6 show that the final run has the best fitting degree with a CC of 0.53 and an RC of 0.71 before position correction. FY-4A, on the other hand, has the worst fitting degree, with a CC of 0.27 and an RC of 0.29. As the position errors of the three SPEs decreased after position correction, the RC and CC between the SPEs and the rain gauge station data samples increase significantly. This means they better fit and correlate with rain gauge data. The regression coefficient of the late run is the highest at this point, at 0.83. FY-4A is the most improved, with a 63% increase in correlation coefficient and a 93.8% increase in regression coefficient.

The relative bias of SPEs with rain gauge stations needs to be studied to verify the improvement of the systematic bias degree of satellite precipitation data by the position correction method. Before position correction, the RB of the final run is the largest and the FY-4A is the smallest, as shown in Figure 7. This is because the final run has the highest precipitation value, which means that if there is a position deviation, the relative bias will be higher; FY-4A has the lowest precipitation value, so even if there is a position bias, the relative bias will be smaller. It also demonstrates that the warping vector field from the automatic registration algorithm significantly impacts the position correction of satellite precipitation estimates. The quantification of the relative bias changes is described with more details in Section B.1.

### 3.3. Error Decomposition

POD, FAR, and CSI are commonly used to quantify the evaluation and analysis of satellite precipitation products. We conducted a graded precipitation test for the three satellite precipitation estimates to investigate the satellite identification of different precipitation intensities. We set the graded test threshold below 15 mm/h due to less than 5% of the frequencies being greater than 15 mm/h at each station in the study area from 19 July to 21 July. Figure 8 shows that as the precipitation threshold is raised, POD and CSI both have a significant downward trend, whereas FAR has a significant upward trend. Before position correction, IMERG’s POD and CSI are higher than FY-4A’s, while FY-4A’s FAR is lower, indicating that IMERG’s satellite precipitation products have better detection accuracy. After position correction, the FAR decreases significantly, the CSI improves, the POD of IMERG remains largely unchanged, and the POD and CSI of FY-4A both show significant improvement. The position correction method has a better improvement effect on FY-4A, as this result once again demonstrates. The quantitative analysis of POD, FAR, and CSI of SPEs are described with more details in Section B.2.

The errors in satellite precipitation estimation result from false alarms, missed reports, and numerical intensity errors of satellite precipitation. Therefore, the total error of SPEs is decomposed into three components: HB, MP, and FP. We calculate the MAE for the TE, HB, MP, and FP to avoid the mutual cancellation of positive and negative errors. It can be found from Table 3 that
(12)$$\begin{array}{c} \mathrm{MAE}(TE)=\mathrm{MAE}(HB)+\mathrm{MAE}(MP)+\mathrm{MAE}(FP). \end{array}$$
The analysis above shows that the position correction method has a significant improvement effect on the error of SPEs.

For the IMERG precipitation product, we find that HB and FP decrease while MP increases. As marked in the lower-left corner of Figure 3, the precipitation area in the southwest of the satellite precipitation becomes smaller after position correction. The area that covered the precipitation of the rain gauge station before becomes uncovered, and similar situations will increase the false alarm error. The reduction of the number of false alarms directly affects the reduction of the false alarm error. Interestingly, Table 3 shows that after position correction, the number of hits of the IMERG precipitation products is reduced, but the hit bias is significantly reduced. The number of hits only represents the ability of the precipitation detected by the satellite, not the accuracy in detecting precipitation. Detecting precipitation in the wrong place, such as data from the precipitation center when it should be detected but detecting other precipitation data around the precipitation center, will increase the hit bias in a large precipitation area. The precipitation center detected by the satellite will be more consistent with the actual position after position correction, and the hit bias will be reduced. The change in total error is consistent with the hit bias, as shown in Figure 9, and the reduction of the hit bias accounts for the majority of the total error reduction contribution. The maximum IMERG total error interval is concentrated between 12:00 and 18:00 on the 20th, and the maximum contribution of hit bias is also concentrated during this time. As the precipitation is at its highest during this time, the hit bias caused by position error is also at its highest.

We found a significant reduction in FP and MP for the FY-4A precipitation product, related to fewer false alarms and missed reports. The number of false alarms has decreased by 44%, making FP reduction the most significant contributor to TE reduction. In extreme precipitation, as shown in Figure 3, the precipitation in the FY-4A inversion is average, and the precipitation center is not visible or distinguishable from the surrounding precipitation. As a result, even though the precipitation hit by the satellite is closer to the actual position, HB is not significantly reduced after position correction. We observe that the reduction of FY-4A precipitation error depends on the decrease in FP from Figure 9, which is consistent with our analysis. Interestingly, we notice that the change in the MP tends to be opposite to the change in the HB, which will cancel each other out in TE. This is consistent with Tang’s study [63].

## 4. Discussion

### 4.1. Effectiveness and Sensitivity of the Method

The registration and warping techniques in image processing are used for position correction of satellite precipitation estimates and the performance of satellite precipitation products of FY-4A and IMERG in the ‘720’ extreme precipitation event in Henan Province is evaluated in this paper. The performance and errors of various satellite precipitation products are compared carefully, and the improvement of satellite precipitation products before and after position correction is investigated thoroughly. According to the experimental results, all satellite precipitation estimates have errors resulting from numerical intensity errors and position bias. Both IMERG products perform similarly to FY-4A and are superior to it. According to the observations from surface rain gauge stations, the final run product, error-corrected with more sensors and monthly average station data from GPCC, has the best numerical intensity. However, its overall performance is inferior to that of the late run after position correction. In extreme precipitation events, FY-4A has difficulty inversely performing light and high-value precipitation, making it difficult for the registration method to obtain image features to generate vector warping fields, resulting in the longest convergence time of its minimization loss function L(p,q,M). The FY-4A product has the highest RMSE, MAE, and average position error compared with the IMERG product, as well as the lowest correlation with station precipitation data and fit. However, FY-4A has the smallest relative bias with concentrated distribution relative to stations due to its little difference in precipitation distribution data. Position correction significantly impacts FY-4A’s performance improvement, and the improvement ratio is the highest of the three, with a 63% improvement in the correlation coefficient and a 93.8% improvement in the regression coefficient, especially in the improvement of the fit to station precipitation.

Additionally, we conducted a sensitivity analysis using the light precipitation event that occurred in Anhui Province, China on 22 March 2022 from 0:00 to 15:00. The hourly average precipitation for this event was less than 0.5 mm/h. The rain gauge data are provided by 67 national surface rainfall stations in Anhui. IMERG late run is selected as the satellite precipitation estimation for testing. Before position correction, the RMSE, MAE, CC, and APE of the IMERG late run were 0.65 mm/h, 0.28 mm/h, 0.62, and 135.87 km, respectively. After position correction, the RMSE, MAE, CC, and APE of the IMERG late run are 0.47 mm/h, 0.19 mm/h, 0.75, and 89.26 km, respectively. The corresponding improvements of RMSE, MAE, CC, and APE are 28%, 32%, 21%, and 34%, respectively. The study shows that the technique still has a positive impact on light precipitation events across various regions, which demonstrates the broad applicability of the method. Combined with Camille’s [44] research, we believe that the density of rainfall stations will affect the registration results, resulting in differences in the improvement effect of the precipitation field, because the position correction method can better capture the characteristics of the precipitation field through the high density of rain gauge stations.

Camille [44] used image registration and warping methods to correct precipitation errors, but there has been no analysis of their error reduction mechanisms. In this paper, an error decomposition model is used to evaluate the error components of satellite precipitation data and investigate how each error component contributes to the total error after position correction. For extreme precipitation events, satellite precipitation covers a large area. Although the IMERG precipitation products have a high POD and CSI, and low FAR, the precipitation intensity values are appropriate and the errors are still significant. This is due to the precipitation centers not being aligned to the proper positions. The IMERG precipitation product primarily benefits from position correction in terms of hit bias. The improvement in hit bias after position correction is not significant for the FY-4A due to its detection inversion mechanism, and it reduces the total error chiefly by decreasing the number of false alarms.

### 4.2. Limitations and Prospects

Although the position correction method through image registration and warping processing can make the SPEs better match the actual precipitation position to reduce the error, it still has some shortcomings. The technique is limited to extracting large-scale features in image processing, focusing on matching the precipitation center but ignoring the surrounding precipitation. As marked in Figure 3, the precipitation distribution from the satellites that initially covered the ground stations may become uncovered after position correction, increasing MP. Image registration requires a certain level of similarity in the input; otherwise, registration failure or a long convergence iteration process may occur. In addition, the method mainly improves SPEs’ position rather than the numerical intensity. This aspect can be improved in the research work in the future.

Satellite remote sensing of a weather system with motion characteristics such as precipitation often has significant position errors limited by sensors and inversion algorithms. This increases the non-Gaussian nature of data errors, affecting climate research and leading to unreliable numerical weather forecasting. In addition, the observation field after position correction can be used as a better initial field for data assimilation and numerical prediction. Therefore, the research on position correction has great significance and broad prospects.

## 5. Conclusions

In this paper, the satellite precipitation products of IMERG and FY-4A during the ‘720’ Henan extreme precipitation event are corrected based on the image registration and warping methods. Then, the error decomposition model is used to study the contribution of each component of different satellite precipitation data errors to the total error after position correction. The main conclusions are given as follows:**Evaluations of SPEs**: Compared with rain gauge stations, the distribution of satellite precipitation estimates in Henan extreme precipitation events has an easterly distribution position error. Among the three satellite precipitation products, IMERG final run has the highest numerical accuracy but FY-4A has the lowest because FY-4A has difficulty inverting very light and very high precipitation values accurately during extreme precipitation events.**Method effects**: The position correction method based on image registration and warping method can effectively reduce the error of satellite precipitation products and FAR, as well as improve POD, CSI, and the fit to the rain gauge stations. The improvement in the final run is limited, with a mean absolute error improvement of only 14%, while the improvement in FY-4A is the most significant, with a mean absolute error improvement of 23% and, in particular, a 63% improvement in its correlation coefficient with rain gauge station data. The improved late run had the best performance and the lowest error. The stability of the position correction method is good and it is also suitable for light precipitation.**Error component improvement contributions**: The IMERG precipitation product primarily benefits from position correction in terms of hit bias, with the biases of hit precipitation improving more than 81%. For the FY-4A, position correction mainly reduces the total error by reducing the number of false alarms, which accounts for 55%, and its improvement on the hit bias is not significant because of the detection inversion mechanisms of the FY-4A.**Limitations**: The registration algorithm is more sensitive to the precipitation center with a high value and will preferentially register the precipitation center. If the precipitation distribution is discontinuous or there are multiple precipitation centers, it may cause instability and failure in the registration process or a long convergence iteration process.

Overall, position corrections for satellite precipitation estimates have some advantages over conventional intensity-only correction techniques. However, the method itself is dependent on the similarity of the two fields. The computational cost of the registration increases with the complexity of the distortion between the fields. The technique can also be used to correct for precipitation as well as weather systems such as hurricanes that have distinct movement characteristics. It is crucial for the application of studies on climate change and weather forecasting to reduce the non-Gaussianity of the geographic data by fusing exact position information with it.

## Figures and Tables

**Figure 1 sensors-22-05583-f001:**
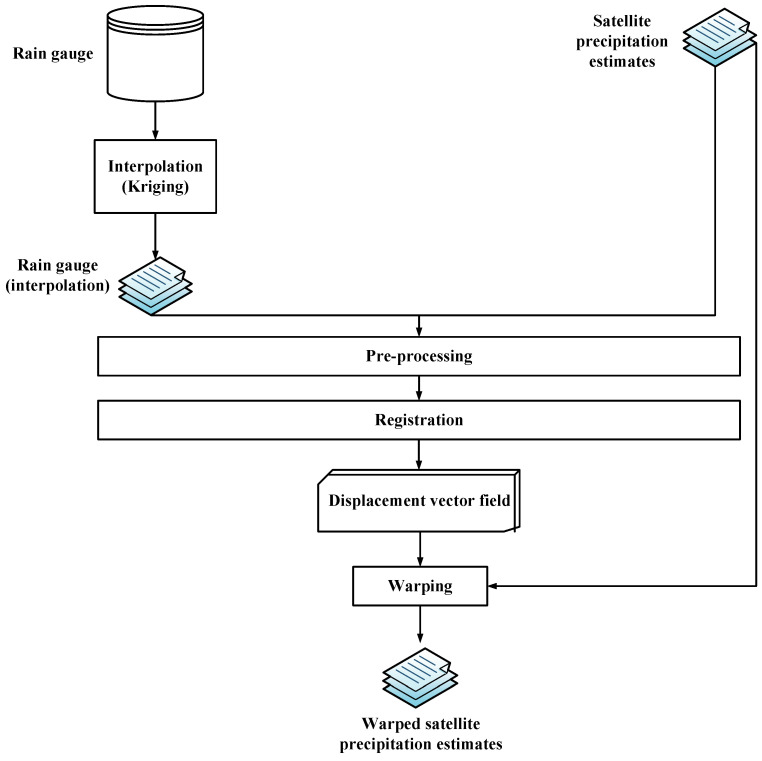
The registration and warping method using rain gauge station data and satellite precipitation estimates.

**Figure 2 sensors-22-05583-f002:**
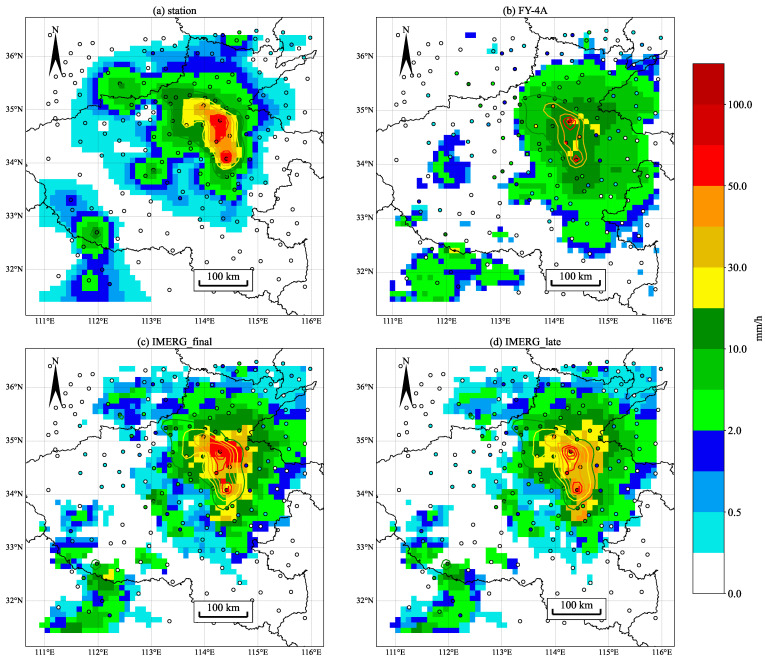
The rain gauge stations (circles) and SPEs (or stations kriged) distribution map at 14:00 on 20 July 2021. (**a**) The precipitation field after interpolating 179 surface rain gauge stations to the satellite grid using kriging interpolation [66,67,68,69]; (**b**–**d**) The precipitation distributions of FY-4A, IMERG final run, and IMERG late run, respectively.

**Figure 3 sensors-22-05583-f003:**
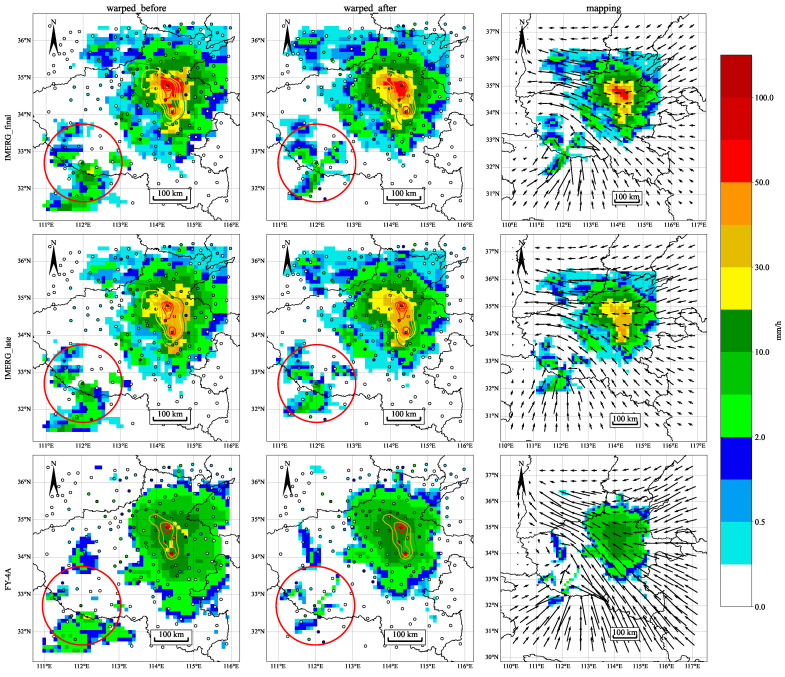
The columns show the distribution of three SPEs before and after image registration, warping processing, and warping vector fields. The rows are IMERG final run, IMERG late run, and FY-4A precipitation products, respectively.

**Figure 4 sensors-22-05583-f004:**
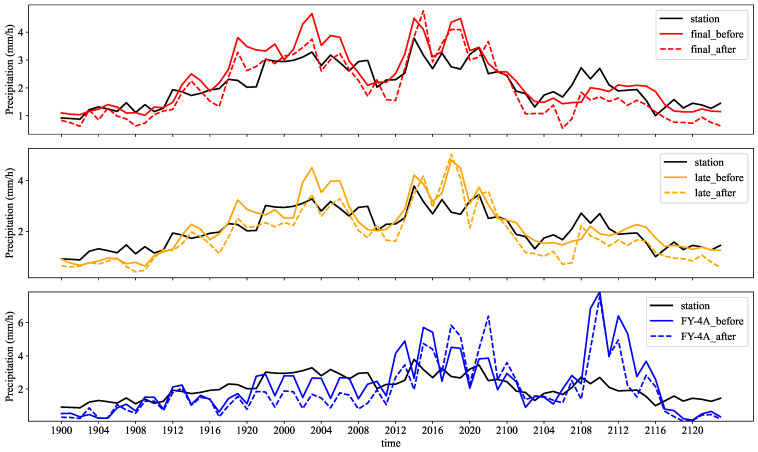
Time series of the average precipitation from 0:00 on 19 July to 23:00 on 21 July. The solid and dashed lines are the time series of average precipitation when the satellite precipitation estimates are interpolated to the station before and after position correction, respectively.

**Figure 5 sensors-22-05583-f005:**
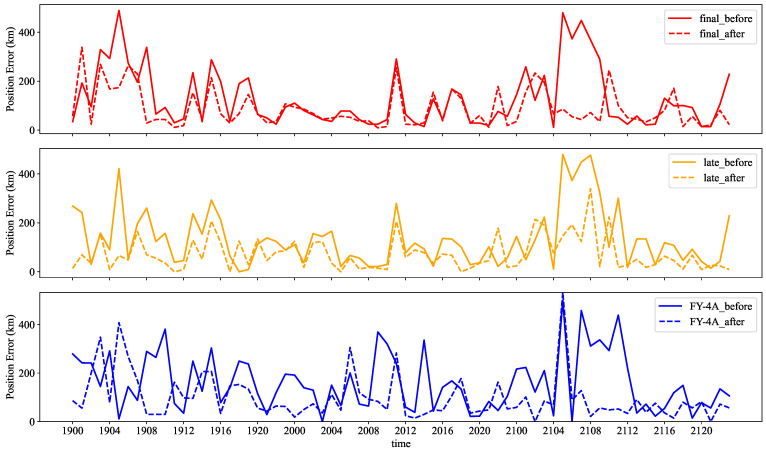
The solid and dashed lines are the time series of the position error before and after position correction from 0:00 on 19 July to 23:00 on 21 July, respectively.

**Figure 6 sensors-22-05583-f006:**
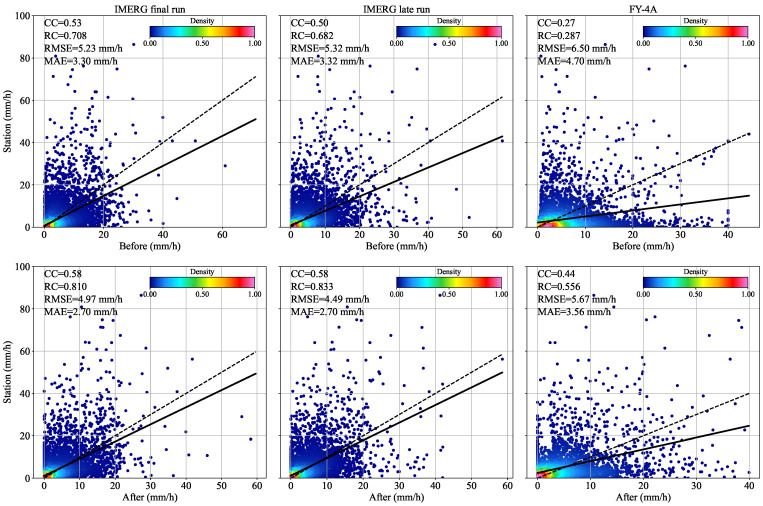
Scatter plots of SPEs and the rain gauge stations before and after position correction. The dashed line is a reference line with a slope of 1, and the solid line is a linear regression fit of the SPEs and the rain gauge stations.

**Figure 7 sensors-22-05583-f007:**
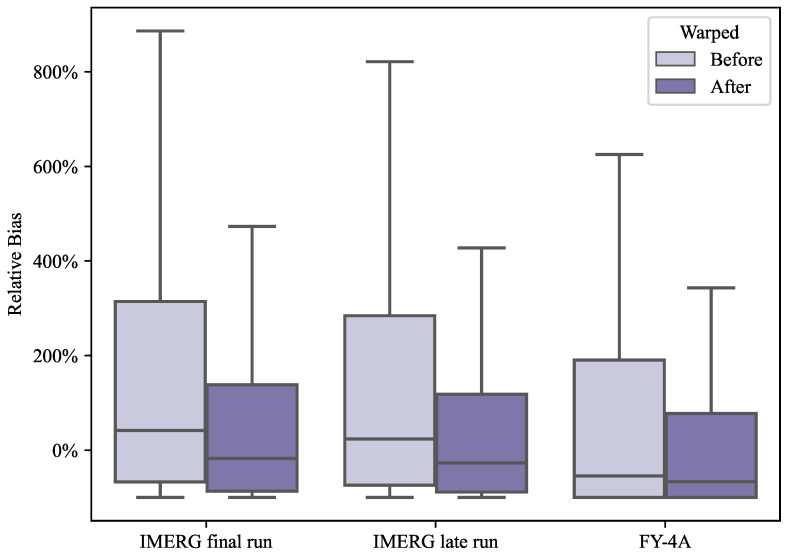
Box plot of relative bias of SPEs with respect to rain gauge stations. The mild outliers are omitted here.

**Figure 8 sensors-22-05583-f008:**
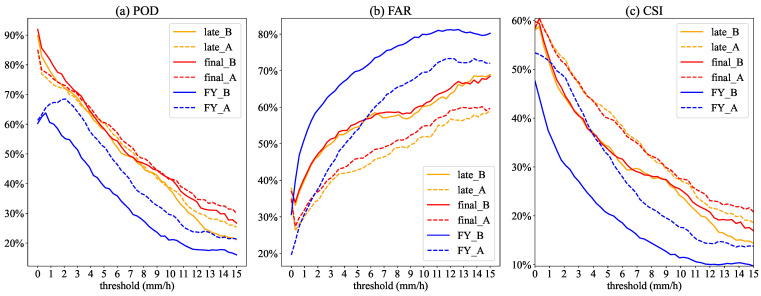
The curves of POD, FAR, and CSI of SPEs before and after correction with thresholds. The solid and dashed lines are the curves before and after position correction, respectively.

**Figure 9 sensors-22-05583-f009:**
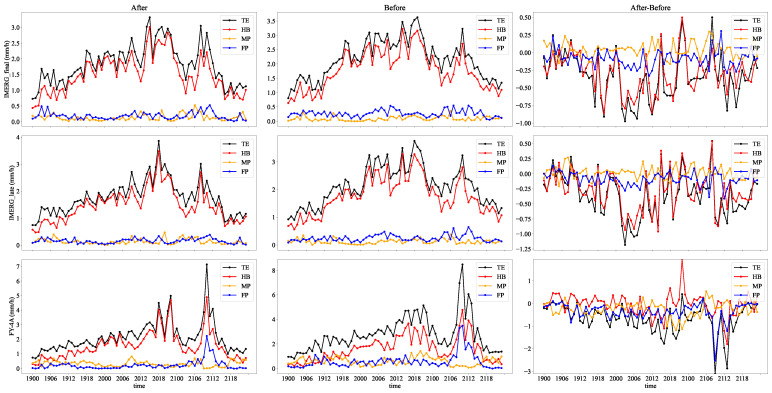
The first two columns show the time series of mean absolute errors of the error components of IMERG final run, IMERG late run, and FY-4A before and after position correction, respectively. The last column is the difference between the absolute mean error of the error components before and after position correction.

**Table 1 sensors-22-05583-t001:** The formula for satellite precipitation evaluation indicators.

Indicators	Formula	Optimal	Units
RMSE	$\sqrt{\frac{1}{n}\sum_{i=1}^{n}{\left(p_{i}-q_{i}\right)}^{2}}$	0	mm/h
MAE	$\frac{1}{n}\sum_{i=1}^{n}\left|p_{i}-q_{i}\right|$	0	mm/h
RB	$\frac{q_{i}-p_{i}}{p_{i}}\times 100\%$	0	%
CC	$\frac{\sum_{i=1}^{n}\left(q_{i}-\bar{q}\right)\left(p_{i}-\bar{p}\right)}{\sqrt{\sum_{i=1}^{n}{\left(q_{i}-\bar{q}\right)}^{2}}\sqrt{\sum_{i=1}^{n}{\left(p_{i}-\bar{p}\right)}^{2}}}$	1	-
POD	$\frac{H}{H+M}$	1	-
FAR	$\frac{F}{H+F}$	0	-
CSI	$\frac{H}{H+F+M}$	1	-

Note: *n* represents the total number of samples. *p* stands for rain gauge station data. *q* represents the SPEs. *H*
represents the number of hits for which the satellite precipitation data captures precipitation events. *F* represents
the number of false alarms for precipitation events from satellite precipitation data. *M* represents the number of
underreported precipitation events from satellite precipitation data.

**Table 2 sensors-22-05583-t002:** Satellite precipitation evaluation indicators before and after position correction.

		RMSE	RMSE_gird	MAE	MAE_grid	CC	CC_grid	APE (km)	RC
final	Before	5.23	3.30	2.15	1.46	0.53	0.61	129.83	0.71
	After	4.97	2.70	1.84	1.08	0.58	0.72	86.74	0.81
late	Before	5.32	3.32	2.12	1.39	0.50	0.58	133.61	0.68
	After	4.49	2.70	1.78	1.03	0.58	0.71	72.17	0.83
FY-4A	Before	6.50	4.70	2.79	2.14	0.27	0.30	161.88	0.29
	After	5.67	3.56	2.13	1.37	0.44	0.55	98.47	0.56

Note: xx_grid represents the error statistic between SPEs and the station after interpolation to the satellite grid.
APE and RC are the average position error and regression coefficient of SPEs and stations, respectively.

**Table 3 sensors-22-05583-t003:** Changes in the mean absolute error of the error components HB, MP, and FP after position correction.

		MAE (TE) mm/h	MAE (HB) mm/h	MAE (MP) mm/h	MAE (FP) mm/h
	Before	2.17	1.81	0.09	0.27
final	After	1.84	1.52	0.13	0.18
	After–Before	−0.34	−0.29	0.04	−0.09
	Before	2.14	1.77	0.12	0.25
late	After	1.78	1.48	0.13	0.16
	After–Before	−0.36	−0.29	0.01	−0.09
	Before	2.79	1.60	0.59	0.59
FY-4A	After	2.13	1.56	0.33	0.23
	After–Before	−0.66	−0.04	−0.26	−0.36

## Data Availability

The rain gauge data in this paper were provided by Dr. Yongzhu Liu from the Center for Earth System Modeling and Prediction, China Meteorological Administration. QPE data of FY-4A were obtained from the China National Satellite Meteorological Center, which can be downloaded from http://satellite.nsmc.org.cn/PortalSite/Data/Satellite.aspx (accessed on 7 June 2022). GPM-IMERG data were obtained from NASA and can be downloaded from https://disc.gsfc.nasa.gov/datasets (accessed on 7 June 2022).

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
