# Peer review of "Technology for Position Correction of Satellite Precipitation and Contributions to Error Reduction—A Case of the ‘720’ Rainstorm in Henan, China"

_sensors, 2022, doi:10.3390/s22155583_

Round 1
Reviewer 1 Report
Please add more explanation about the precipitation event ’720’ in Henan Province. With a better characterization of the testing event, the merit of the proposed method could be better appreciated.
Another issue is that the characteristics of the test case for the method were not sufficiently explained. Can the correction developed for this extreme rainfall event be applied to other rainfall events? In other words, discussions about the range of applicability will be more useful to readers.
Author Response
Dear reviewer:
Thank you for your valuable advice. We have completed manuscript revisions. Please check the attachment.

Reviewer 2 Report
Generally speaking, I appreciate the author's contribution. Manuscript has caught the hot topic and tried to explore the method of Correct satellite precipitation data, which is of great significance. However, it is suggested that the author make key improvements in the following aspects.
1. The “Introduction” needs to be strengthened. The theoretical background is insufficient, and there are too few literature covering related fields, and many statements are not supported by enough literature. There is a lack of summary of existing research.
2. In terms of data source, the surface rain gauge data is very important, so please provide more detailed information
3. The author should reorganize the introduction of the method so that more readers can understand it.
4. As far as I know, Kriging is not the only interpolation method. Is it the most suitable one in the text?
5. Please explain the basis and rationality of the selection of precipitation Satellite Data Evaluation Indicators.
6. The overall length of the “Results and analysis “part is too long, so it is suggested to focus on the core result.
7. Is it necessary for each chart to be shown in the text? Can some be presented as appendices? Some graphics are not standardized and cannot be read independently of the text. The map lacks scale.
8. The “Discussion” lacks comparison with similar studies.
9. The “Conclusion” should be a further refinement of the results.
Author Response

(The authors gave the same response as above.)

Round 2
Reviewer 2 Report
Thanks to the author for the revision, but before the publication of the article, some aspects still need to be improved.
(1) Language and normative issues. There were many new contents added in the revised manuscript, but grammar and logical cohesion should also be improved. Many statements are not concise enough. The citation format of references and materials in this paper is still not standard, and there are some minor errors. Please have a comprehensive inspection.
(2) The "Abstract" needs to be adjusted to correspond to the revised content.
(3) It is suggested to add secondary titles to the "discussion" to enhance readability.
(4) The "Conclusion" needs to be further summarized, improved, and simplified.
Author Response
Dear reviewer:
Thank you for your valuable advice. We have completed manuscript revisions. Please check the attachment.

This manuscript is a resubmission of an earlier submission. The following is a list of the peer review reports and author responses from that submission.